neuroscience

neuromuscular junction, NMJ-morph, macro, ImageJ, Fiji

**Author for correspondence:**
Ross A. Jones
e-mail: ross.jones@ed.ac.uk

†These authors contributed equally to this study.

# aNMJ-morph: a simple macro for rapid analysis of neuromuscular junction morphology

Gavin Minty[1,2,†], Alex Hoppen[3,†], Ines Boehm[1,2,†], Abrar Alhindi[1,2], Larissa Gibb[1,2], Ellie Potter[1,2], Boris C. Wagner[1,2], Janice Miller[4], Richard J. E. Skipworth[4], Thomas H. Gillingwater[1,2] and Ross A. Jones[1,2]

[1]Edinburgh Medical School: Biomedical Sciences (Anatomy), and [2]Euan MacDonald Centre for Motor Neurone Disease Research, University of Edinburgh, Edinburgh EH8 9AG, UK
[3]RWTH Aachen University, 52062 Aachen, Germany
[4]Clinical Surgery, University of Edinburgh, Royal Infirmary of Edinburgh, Edinburgh, EH16 4SA, UK

IB, 0000-0002-9343-0944; THG, 0000-0002-0306-5577; RAJ, 0000-0002-1569-2500

Large-scale data analysis of synaptic morphology is becoming increasingly important to the field of neurobiological research (e.g. 'connectomics'). In particular, a detailed knowledge of neuromuscular junction (NMJ) morphology has proven to be important for understanding the form and function of synapses in both health and disease. The recent introduction of a standardized approach to the morphometric analysis of the NMJ—'NMJ-morph'—has provided the first common software platform with which to analyse and integrate NMJ data from different research laboratories. Here, we describe the design and development of a novel macro—'automated NMJ-morph' or 'aNMJ-morph'—to update and streamline the original NMJ-morph methodology. ImageJ macro language was used to encode the complete NMJ-morph workflow into seven navigation windows that generate robust data for 19 individual pre-/post-synaptic variables. The aNMJ-morph scripting was first validated against reference data generated by the parent workflow to confirm data reproducibility. aNMJ-morph was then compared with the parent workflow in large-scale data analysis of original NMJ images (240 NMJs) by multiple independent investigators. aNMJ-morph conferred a fourfold increase in data acquisition rate compared with the parent workflow, with average analysis times reduced to approximately 1 min per NMJ. Strong concordance was demonstrated between the two approaches for all 19 morphological variables, confirming

the robust nature of aNMJ-morph. aNMJ-morph is a freely available and easy-to-use macro for the rapid and robust analysis of NMJ morphology and offers significant improvements in data acquisition and learning curve compared to the original NMJ-morph workflow.

# 1. Background

Synaptic connectivity is central to the structure and functioning of the mammalian nervous system. In practice, the detailed analysis of synaptic connectivity—'connectomics'—remains a formidable task. Even in small laboratory animals (e.g. mice, rats) that are routinely used to model human disease, the cerebral cortex may contain up to 1700 synaptic connections per 1500 µm$^3$ volume [1]. Given the complexity of the central nervous system, the study of 'model synapses' has been critical to the progress of synaptic biology, with the neuromuscular junction (NMJ)—the synapse between lower motor neuron and skeletal muscle fibre—representing the paradigm example.

The importance of normal synaptic connectivity is evidenced by the multitude of neurodegenerative conditions that are underpinned by synaptic dysfunction and/or degeneration at the NMJ. For example, myasthenia gravis and its related syndromes, along with motor neuron diseases such as amyotrophic lateral sclerosis and spinal muscular atrophy, all demonstrate varying degrees of synaptic pathology at the NMJ as either a cause or consequence of the underlying disease mechanism [2–5]. Finding effective treatments for these conditions ultimately depends on a greater understanding of the normal and pathological architecture of mammalian synapses, including NMJs.

Until recently, however, even basic quantification of the gross cellular anatomy of the NMJ has been hampered by the lack of a standardized approach to morphometric analysis. Following the introduction of NMJ-morph in 2016 [6]—a simple but robust method for NMJ quantification—a growing number of research groups have now used this approach to gain important insights into a diverse range of conditions and species [7–12]. For example, NMJ-morph was pivotal to the first major study on the cellular and molecular anatomy of the human NMJ [7]. Here, NMJ-morph revealed in detail the unique 'nummular' morphology of the human NMJ and further demonstrated its structural stability over the lifespan, in direct contrast to age-related fragmentation of rodent NMJs. The sensitivity of NMJ-morph analysis has identified subtle changes in NMJ morphology found in Charcot–Marie–Tooth disease [8] and helped characterize NMJ degeneration in CHCHD10-encoded mitochondrial myopathy associated with motor neuron disease [9]. Most recently, NMJ-morph has been used in the study of human pathology, revealing that NMJs are stable in patients with cancer cachexia—the severe loss of skeletal muscle that is commonly associated with many forms of cancer [12].

At present, the two major barriers to more widespread adoption of NMJ-morph are its associated learning curve, and relatively low data throughput in real time (approx. 12 NMJs h$^{-1}$). Here, we present a macro update to the original NMJ-morph workflow—'automated NMJ-morph' or 'aNMJ-morph'—to streamline and expedite data acquisition.

# 2. Results and discussion

To support the continued uptake of NMJ-morph in the field of synaptic biology and related disciplines, we developed a macro update of the original workflow—'aNMJ-morph'. The full version of the macro (compatible with both Windows and Mac operating systems) and supporting materials (including tutorial videos, sample images and reference spreadsheets) is freely available for download at Edinburgh DataShare [13]. For a complete understanding of the individual morphological variables (and their derivations), we recommend that users of aNMJ-morph are familiar and competent with the use of the original workflow in a practical setting [6].

The standard NMJ-morph workflow uses ImageJ/Fiji [14] and the Binary Connectivity [15] plugin (all in the public domain) to generate data for 19 different morphological variables on confocal images of individual NMJs [6]. For each image, this workflow requires the user to navigate through approximately 75 separate drop-down menus in Fiji, followed by manual input of raw data into a spreadsheet proforma (containing formulae for generating additional derived variables). The initial published estimate of throughput suggested a work rate of approximately 30 NMJs h$^{-1}$ for an experienced user [6]; in practice, we have found that most users are able to analyse approximately 12 NMJs h$^{-1}$ (approx. 5 min per NMJ), though

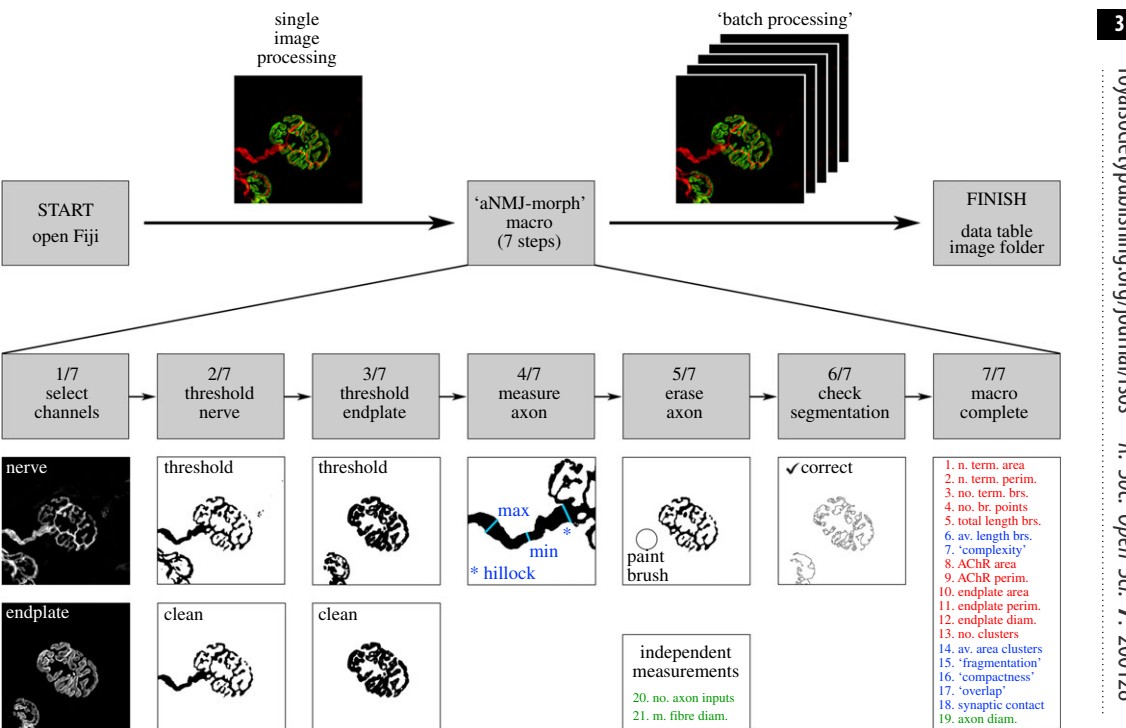

**Figure 1.** 'aNMJ-morph'. The 'aNMJ-morph' macro comprises seven instruction windows that guide the user through the various stages of image analysis, and can be used for either single image or batch processing. The only manual inputs include image thresholding, axon processing (measure/erase) and confirmation of image segmentation. At completion, aNMJ-morph generates a data table containing 19 individual morphological variables, corresponding to those of the original NMJ-morph workflow; the 'number of axonal inputs' and 'muscle fibre diameter' are measured independently. 'Core variables' are shown in red typeface, 'derived variables' in blue and 'associated nerve and muscle variables' in green. Note: For single NMJ analysis, first open the image, then select the macro from the plugins. For batch processing, first open the macro, then select the image folder; the macro will automatically cycle through each image in turn to completion.

the throughput can be increased by the use of keyboard shortcuts built-in to ImageJ, and by the assignment of additional shortcuts.

In comparison, aNMJ-morph (figure 1) streamlines the workflow into seven instruction windows and tabulates the results automatically (in the form of a .csv spreadsheet), reducing the image acquisition time to just under 1.5 min per NMJ. In addition to the substantial time saving, one of the major advantages of aNMJ-morph is the simplicity of data acquisition. This enables the user to focus on the most critical step—accurate image thresholding—and reduces common errors resulting from data transcription and transposition [6].

To screen for any unanticipated scripting and/or technical issues arising during the development of the macro, a reference dataset (of 40 NMJs) was first analysed by a single investigator using both the original (NMJ-morph) and automated (aNMJ-morph) workflows. For this exercise, the same threshold settings were selected for both manual and automated assessments. As expected, the results of this initial validation generated near perfect correlations ($r = 0.954$–$1.000$; all $p < 0.0001$; table 1).

The fractionally lower correlation coefficients for the pre-synaptic variables ($r = 0.954$–$0.998$; table 1) compared to the post-synaptic variables are a consequence of the manual erasing of nerve terminal axon following axonal diameter measurement. Manual input also accounted for the minor differences in endplate diameter between the two methods ($r = 0.992$; table 1 and figure 2e). For the number of AChR clusters ($r = 0.986$; table 1) and its derivations (fragmentation and average area of AChR clusters), the discrepancy between NMJ-morph and aNMJ-morph was methodological (figure 2d). In 3 of 40 endplates, the 'fill holes' function in the macro reduced the total number of clusters in each NMJ by one, due to the enclosure of a single AChR cluster within another on the segmented image (figure 2d). These occasional examples were not found to have any statistically significant effect in practice (see below).

To assess the usability of aNMJ-morph in practice, two pairs of investigators were then tasked with analysing a large volume of new NMJ images (figure 3 and table 1) using either the original NMJ-morph

**Table 1.** NMJ-morph (manual) versus aNMJ-morph (macro). Correlation coefficients ($r$) comparing the two methods of image analysis for each variable. During the development of aNMJ-morph, a single investigator applied the two approaches using the same threshold settings ($n = 40$ NMJs; *Within User*). After validation, two pairs of investigators worked in real time on a large image bank using either aNMJ-morph or the original workflow ($n = 240$ NMJs; *Between User*). Correlation coefficients support the robust nature of the aNMJ-morph macro in a practical setting. Correlation coefficients ($r$) are Pearson for parametric variables, Spearman for non-parametric variables; $p < 0.0001$ for all correlation coefficients.

| | NMJ-morph (manual) versus aNMJ-morph (macro) | |
| --- | --- | --- |
| morphological variable | *Within User (r)* | *Between User (r)* |
| pre-synaptic | | |
| (1) nerve terminal area ($\mu m^2$) | 0.998 | 0.892 |
| (2) nerve terminal perimeter ($\mu m$) | 0.994 | 0.875 |
| (3) number of terminal branches | 0.978 | 0.762 |
| (4) number of branch points | 0.987 | 0.740 |
| (5) total length of branches ($\mu m$) | 0.977 | 0.791 |
| (6) average length of branches ($\mu m$) | 0.954 | 0.661 |
| (7) 'complexity' | 0.978 | 0.785 |
| post-synaptic | | |
| (8) AChR area ($\mu m^2$) | 1.000 | 0.923 |
| (9) AChR perimeter ($\mu m$) | 1.000 | 0.858 |
| (10) endplate area ($\mu m^2$) | 1.000 | 0.982 |
| (11) endplate perimeter ($\mu m$) | 1.000 | 0.949 |
| (12) endplate diameter ($\mu m$) | 0.992 | 0.891 |
| (13) number of AChR clusters | 0.986 | 0.937 |
| (14) average area of AChR clusters ($\mu m^2$) | 0.971 | 0.823 |
| (15) 'fragmentation' | 0.986 | 0.936 |
| (16) 'compactness' (%) | 1.000 | 0.827 |
| (17) 'overlap' (%) | 1.000 | 0.765 |
| (18) area of synaptic contact ($\mu m^2$) | 1.000 | 0.914 |
| associated nerve and muscle | | |
| (19) axon diameter ($\mu m$) | 0.960 | 0.758 |

workflow or the macro. Images were obtained from ongoing research projects and included NMJs from a range of both slow and fast-twitch muscles (e.g. soleus and extensor digitorum longus, respectively; $n = 240$ NMJs in total). In addition, the new images were of a different file format (.nd2, Nikon) to those used for the initial macro development (.lsm, Zeiss; see Methods) and each investigator used a different workstation and operating system (to ensure compatibility with both Windows and Mac).

As before, correlation analyses revealed strong concordance between the two approaches (NMJ-morph versus aNMJ-morph) for all variables ($r = 0.661$–$0.982$; all $p < 0.0001$; table 1). The greater range of correlation coefficients highlights the normal inter-user variability that is expected in relation to thresholding and manual data input, and is in keeping with the variability described in the original NMJ-morph workflow [6]. Crucially, in relation to the counting of AChR clusters, correlations were strong between the two methods ($r = 0.937$, $p < 0.0001$; table 1), supporting this approach in the automation process (figure 2d).

Of particular note, aNMJ-morph conferred a fourfold increase in data acquisition rate, with average analysis time per NMJ reduced from nearly 5.5 min (319 s) to just over 1 min (79 s). In practical terms, this represents a substantial improvement in work rate and potential throughput. To enable robust comparison of different NMJ populations (e.g. muscles, animals, species, etc.) we recommend datasets of at least 30–40 NMJs per sample based on NMJ-morph guidelines [6]; in real time, complete NMJ datasets can now be obtained in just over half an hour with aNMJ-morph, compared to around

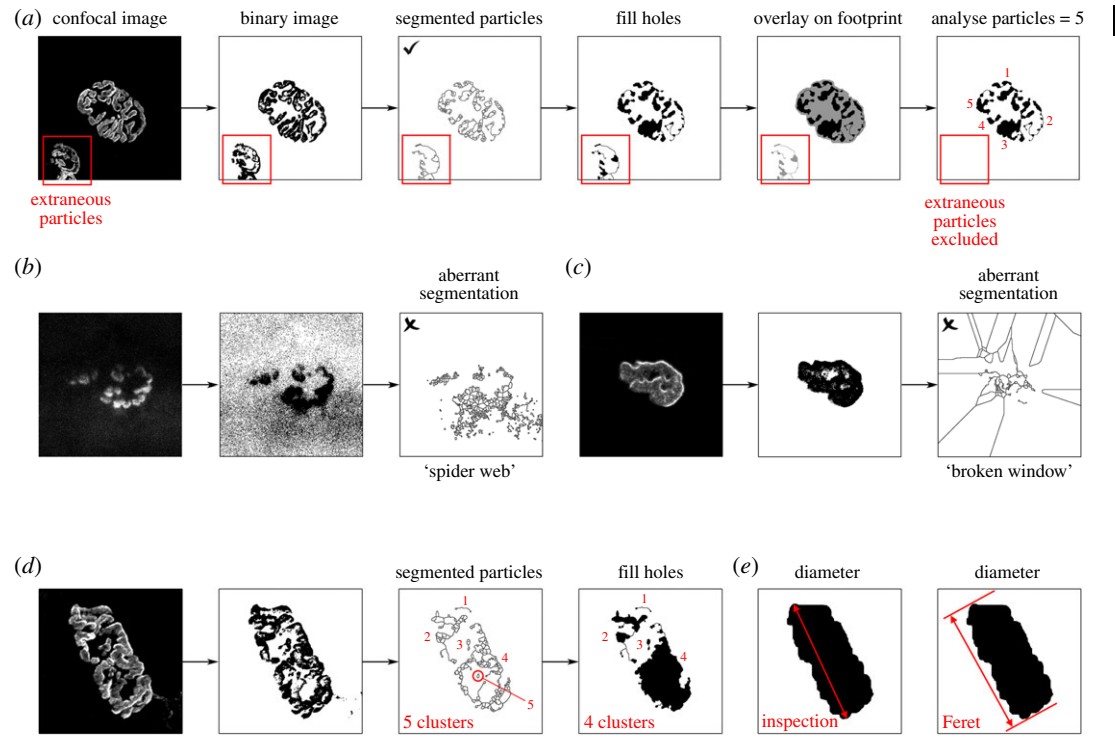

**Figure 2.** Automation within aNMJ-morph. Several processes within the original NMJ-morph workflow required additional scripting to enable full automation. (*a*) Automated counting of AChR clusters necessitated the exclusion of extraneous background particles. (*b,c*) Examples of aberrant image segmentation. These images are identified at the 'check segmentation' step of aNMJ-morph (window 6/7; figure 1). (*d*) Variation in particle number between NMJ-morph and aNMJ-morph (in this example, five clusters versus four clusters); in practice, these occasional examples of spurious counting were not found to be statistically significant. (*e*) Automation of endplate diameter measurement using the Feret's diameter function in ImageJ/Fiji.

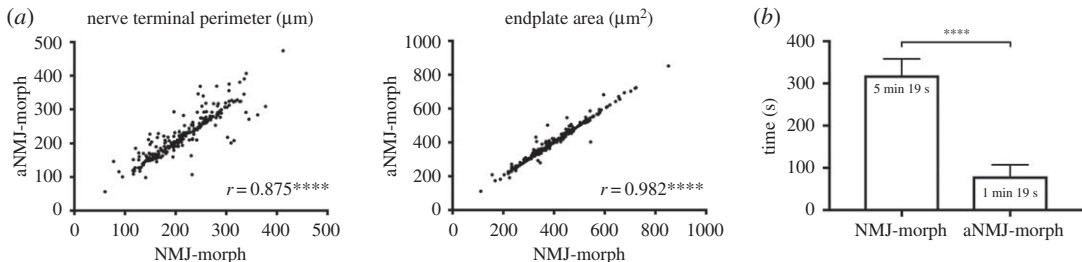

**Figure 3.** NMJ-morph (manual) versus aNMJ-morph (macro). aNMJ-morph offers a robust and expeditious alternative to the original NMJ-morph workflow. Two pairs of investigators analysed a large image bank ($n = 240$ NMJs) using either aNMJ-morph or the original workflow. (*a*) Correlation analyses demonstrated strong concordance between the two methods for all variables; examples of pre- and post-synaptic variables are illustrated (nerve terminal perimeter and endplate area). (*b*) aNMJ-morph conferred a fourfold reduction in analysis time (approx. 1 min per image) compared with the original workflow (approx. 5 min per image). Pearson correlation; ****$p < 0.0001$.

3 h or so previously, depending on the level of proficiency with NMJ-morph. In addition, aNMJ-morph eliminates the common errors associated with manual data transfer via an automatically curated .csv file containing the 19 morphometric variables (and additional information on image size and threshold selection).

We anticipate that other research groups will now wish to trial the macro in different settings, e.g. with NMJ images acquired using different scanning parameters and/or file types. To support these adaptations, we recommend that users first validate the macro output against equivalent data generated using the original workflow [6] to confirm the functionality of the macro in different settings. We also encourage the development of machine-learning algorithms based on the existing NMJ-morph approach to further refine and improve the rate of data acquisition.

# 3. Conclusion

'Automated NMJ-morph'—'aNMJ-morph'—is a freely available update to the existing NMJ-morph workflow, for the rapid and robust analysis of NMJ morphology. aNMJ-morph offers significant advantages over the original workflow, with a clear emphasis on accessibility and ease of use. It is hoped that aNMJ-morph will be of particular interest to NMJ biologists and associated researchers who are engaged in large-scale data analysis of comparative NMJ morphology.

# 4. Methods

## 4.1. Macro scripting and validation

A Fiji/ImageJ-based macro (automated NMJ-morph or aNMJ-morph) was first scripted using ImageJ macro language (IJM) [16] to encode the complete NMJ-morph workflow as described in the original manuscript [6]. The full IJM-text transcription is included in electronic supplementary material, file S1. The final aNMJ-morph macro comprises seven instruction windows and generates a spreadsheet containing data for 19 individual pre-/post-synaptic variables (figure 1).

For the majority of operations in the NMJ-morph workflow, the IJM scripting involved straightforward coding of the correct sequence of drop-down menus and checkbox selections within Fiji. Several operations required further development to enable full automation, including the 'number of AChR clusters' and 'endplate diameter' (figure 2).

Assessment of the 'number of AChR clusters' in the original NMJ-morph workflow involved manual counting of 'segmented particles' (figure 2a, panels 1–3) in order to distinguish genuine clusters (i.e. those contributing to the endplate) from extraneous particles (e.g. adjacent endplates or background noise). For aNMJ-morph, we were able to automate this process via several additional steps in the macro scripting (figure 2a, panels 4–6; electronic supplementary material, file S1). These steps involved overlaying filled particles onto the footprint of the endplate (using the 'fill holes' and 'concatenate' functions). Automated counting (analyse particles) then returned the number of particles lying within the footprint alone (i.e. those contributing to the endplate) while excluding any extraneous particles.

On rare occasions (less than 1 in 1000 images), the use of the 'segmented particles' function in the original NMJ-morph workflow resulted in spurious fragmentation of the endplate, with images resembling 'spider webs' or 'broken windows' (figures 2b,c). In our experience, this was usually the result of poor image quality from the outset (figure 2b); as per the original guidelines [6], we recommend that these NMJs are excluded entirely. In exceptional circumstances, aberrant segmentation is noted in images of sound quality (figure 2c); in these instances, automated counting of clusters is not possible (measurement of other variables is unaffected, e.g. area, perimeter, etc.). To address these eventualities in the macro, an instruction window was incorporated prompting users to confirm appropriate segmentation of the image (figure 1, window 6/7; electronic supplementary material, file S1); in circumstances of abnormal segmentation, the macro will still measure and record the other variables.

The measurement of 'endplate diameter' was the only other variable that required automation. In the original NMJ-morph workflow, the maximum linear dimension of the endplate was judged on inspection and recorded manually. In the macro, this value was obtained automatically by using the 'Feret's diameter' function in ImageJ, which provides an analogous measurement (figure 2e; electronic supplementary material, file S1).

The only manual aspects of the original NMJ-morph workflow to be retained in the macro related to image thresholding and axon processing (figure 1, windows 1–5/7). Accurate image thresholding is critical to the generation of robust NMJ-morph data [6] and it was crucial to retain this step under user-defined control; the original NMJ-morph manuscript [6] should be consulted for detailed instruction/discussion of image thresholding. Of note, thresholded binary images must be compared to the original raw images to confirm accurate image reproduction. Similarly, the measurement of axon diameter requires a degree of user-dependent decision-making that is not compatible with simple automation, particularly in relation to NMJs of certain species, e.g. human NMJs [7,12].

Two further variables are conventionally recorded as part of a complete NMJ-morph analysis—'number of axonal inputs' and 'muscle fibre diameter'. Since both variables require independent measurement, they were not suitable for automation. Polyinnervation (i.e. number of axonal inputs > 1) only occurs in certain specific circumstances (e.g. development, pathology) and requires careful assessment, while muscle fibre diameter is measured on a separate set of images [6].

During development, aNMJ-morph was compared against the original workflow by a single investigator using the same image threshold settings. To assess the usability of aNMJ-morph in a wider context (different investigators, different laboratories, etc.), four different investigators trialled the two methods on a much larger image bank (see Results).

## 4.2. NMJ images and file types

All NMJ images used in the development and testing of the aNMJ-morph macro were obtained from previous and/or ongoing animal research projects covered by the requisite personal and project licences granted by the UK Home Office. All images were captured using Zeiss/Nikon confocal microscopes, with the file types .lsm/.nd2, respectively. The macro uses the maximum intensity projection of the corresponding *z*-stack and has been validated for these file types and image formats only. For all other file types, we recommend that users first validate the macro output against equivalent data generated with the original workflow [6] before proceeding.

## 4.3. ImageJ/Fiji and binary connectivity

The macro was developed using ImageJ/Fiji software (v.: 2.0.0-rc-67/1.52i / build: 1762a07c5c). The latest version of ImageJ/Fiji is freely available at https://fiji.sc [14] including instructions for download. The Binary Connectivity plugin is freely available at https://blog.bham.ac.uk/intellimic/g-landini-software [15] under the section 'Morphological Operators for ImageJ' (including instructions for installation). To manage updates, the latest version of the macro will be hosted at Edinburgh DataShare [13].

## 4.4. Statistical analysis

All statistical analyses were performed on GraphPad Prism software; individual statistical tests are indicated in the relevant figure legends.

Data accessibility. The aNMJ-morph macro, tutorial video, sample images and reference spreadsheets are available for download at Edinburgh DataShare: https://doi.org/10.7488/ds/2625 [13].

Authors' contributions. G.M., A.H., I.B. and R.A.J. developed the aNMJ-morph macro. G.M., A.H., I.B., A.A., L.G., E.P., B.C.W., T.H.G. and R.A.J. performed all experiments and data analysis. All authors drafted and approved the manuscript for submission.

Competing interests. The authors have no competing interests to declare.

Funding. This work was supported by funding from Biomedical Sciences (Anatomy) at the University of Edinburgh (T.H.G., R.A.J.) and a small research pump priming grant from the Royal College of Surgeons of Edinburgh (J.M., R.J.E.S., R.A.J.). R.J.E.S. is also supported by an NHS Research Scotland (NRS) clinician post. J.M. is also supported by Cancer Research UK.

Acknowledgements. The authors are particularly grateful to Matthew Robinson for his helpful suggestions on the early development of the macro.

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
