## [Reviewer comments · Royal Society Open Science]

Review History

RSOS-200128.R0 (Original submission)

Review form: Reviewer 1

Is the manuscript scientifically sound in its present form?

Yes

Are the interpretations and conclusions justified by the results?

Yes

Is the language acceptable?

Yes

Do you have any ethical concerns with this paper?

No

Have you any concerns about statistical analyses in this paper?

No

Recommendation?

Accept with minor revision (please list in comments)

Comments to the Author(s)

This, aNMJ-Morph macro, is an important improvement of an existing method to increase the time and congruence of analyzing NMJs. Despite the improvements, there is little mention of how the macro would perform with images of NMJs acquired using different scanning parameters, a critical consideration. There is also little mention if machine-learning algorithms could potentially outperform this macro and importantly be a more reliable and speedier method for assessing the morphology of NMJs taken using scanning parameters. Thus, the discussion should at least elaborate on this potential issues and future improvements.

Review form: Reviewer 2**Is the manuscript scientifically sound in its present form?**

Yes

Are the interpretations and conclusions justified by the results?

No

Is the language acceptable?

Yes

Do you have any ethical concerns with this paper?

No

Have you any concerns about statistical analyses in this paper?

No

Recommendation?

Major revision is needed (please make suggestions in comments)

Comments to the Author(s)

The manuscript RSOS-200128 by Gavin Minty et al. describes the novel macro for semi-automated morphometric analysis of the confocal microscopy images of neuromuscular junctions. This macro-based analysis is a development of the manual workflow that this group published in 2016. The novel macro seems useful for the field, and the authors are making the software freely available. The manuscript is interesting and seems suitable for this journal; however, several points need to be addressed. Detailed review critiques are described below in the order of appearance and not by importance.

Page 3, lines 31 to 35, "aNMJ-morph conferred a 5-fold increase in data acquisition rate compared with the parent workflow, with average analysis times reduced to approximately 1 minute per NMJ" is an over-interpretation and needs revision.

- Page 7, lines 19 to 22, "average analysis time per NMJ reduced from nearly 5 and a half minutes (319 seconds) to just over 1 minute (79 seconds)." This improvement is only four-fold ($319/79=4.04$).
- The same issue is seen on page 14, lines 54 to 58 ($319/79=4.04$).
- The same issue applies to page 6, line 3, "the image acquisition time to \square 1 minute per NMJ" seems like an overstatement of the difference.

Page 5, line 38.

Instructions should be given where to obtain the required plugin "the Binary Connectivity plugin" and how to install it.

Page 6, lines 53-57.

Has the macro been tested in both Windows or Macintosh platforms? If not, authors should specify which operating system has been used for testing the macro.

Page 8, Methods.

Describe the Fiji version number, the Fiji build number, and the manage update sites that are necessary to execute this workflow.

Page 8, lines 52 to 57. The Reviewer agrees to the authors for figure 2B being poor image quality. However, for figure 2C, the thresholding has been appropriately executed without evident noise in the image. Thus, the Reviewer disagrees with concluding this analysis problem as a result of poor image quality. The authors need to investigate the cause further and how to deal with this kind of situation. Most importantly, the Reviewer disagrees with excluding NMJs like this image from the analysis.

Page 9, lines 23 to 25.

The authors should describe in detail about thresholding, whether it is appropriate to use the same or different thresholding for each NMJ image.

Page 9, line 38, "the were" should be corrected.

Page 10, lines 3 to 9, "NMJ images."

The authors must elaborate on the images used in this macro. What image file-type would be acceptable for the analysis? Which microscope manufacture original file type could be used directly in this workflow? Do the users need to load additional Fiji plugins to read specific file types necessary for the aNMJ-morph?

If not, specify which file type is compatible. The Reviewer assumes that the confocal Z-stack needs to be projected. If so, specify what type of projection is suitable for this analysis?

Page 14, line 11, "batch processing." Instruction seems to be missing for how to batch process images.

Page 14, lines 12 to 14, "The Reviewer assumes that the confocal Z-stack needs to be projected. If so, specify what type of projection is suitable for this analysis?"

Erasing the axon using a paintbrush is a manual input to the analysis. The same issue is seen on page 9, lines 19 to 23.

Review form: Reviewer 3

Is the manuscript scientifically sound in its present form?

Yes

Are the interpretations and conclusions justified by the results?

Yes

Is the language acceptable?

Yes

Do you have any ethical concerns with this paper?

No

Have you any concerns about statistical analyses in this paper?

No

Recommendation?

Accept with minor revision (please list in comments)

Comments to the Author(s)

See attached file.

Decision letter (RSOS-200128.R0)

16-Mar-2020

Dear Dr Jones

On behalf of the Editors, I am pleased to inform you that your Manuscript RSOS-200128 entitled "aNMJ-morph – A simple macro for rapid analysis of neuromuscular junction (NMJ) morphology" has been accepted for publication in Royal Society Open Science subject to minor revision in accordance with the referee suggestions. Please find the referees' comments at the end of this email.

The reviewers and handling editors have recommended publication, but also suggest some minor revisions to your manuscript. Therefore, I invite you to respond to the comments and revise your manuscript.

- Ethics statement

- Data accessibility

<http://datadryad.org/submit?journalID=RSOS&manu=RSOS-200128>

- Competing interests

- Authors' contributions

All submissions, other than those with a single author, must include an Authors' Contributions section which individually lists the specific contribution of each author. The list of Authors

should meet all of the following criteria; 1) substantial contributions to conception and design, or acquisition of data, or analysis and interpretation of data; 2) drafting the article or revising it critically for important intellectual content; and 3) final approval of the version to be published.

- Acknowledgements

- Funding statement

Because the schedule for publication is very tight, it is a condition of publication that you submit the revised version of your manuscript before 25-Mar-2020. Please note that the revision deadline will expire at 00.00am on this date. If you do not think you will be able to meet this date please let me know immediately.

- 1) A text file of the manuscript (tex, txt, rtf, docx or doc), references, tables (including captions) and figure captions. Do not upload a PDF as your "Main Document";
- 2) A separate electronic file of each figure (EPS or print-quality PDF preferred (either format should be produced directly from original creation package), or original software format);

- 3) Included a 100 word media summary of your paper when requested at submission. Please ensure you have entered correct contact details (email, institution and telephone) in your user account;
- 4) Included the raw data to support the claims made in your paper. You can either include your data as electronic supplementary material or upload to a repository and include the relevant doi within your manuscript. Make sure it is clear in your data accessibility statement how the data can be accessed;
- 5) All supplementary materials accompanying an accepted article will be treated as in their final form. Note that the Royal Society will neither edit nor typeset supplementary material and it will be hosted as provided. Please ensure that the supplementary material includes the paper details where possible (authors, article title, journal name).

If your manuscript is newly submitted and subsequently accepted for publication, you will be asked to pay the article processing charge, unless you request a waiver and this is approved by Royal Society Publishing. You can find out more about the charges at <https://royalsocietypublishing.org/rsos/charges>. Should you have any queries, please contact openscience@royalsociety.org.

on behalf of Dr Sean Murray (Associate Editor) and Malcolm White (Subject Editor)
openscience@royalsociety.org

Associate Editor Comments to Author (Dr Sean Murray):
Comments to the Author:

Dear Dr Jones,

Thank you for submitting your manuscript 'aNMJ-morph – A simple macro for rapid analysis of neuromuscular junction (NMJ) morphology' for consideration at Royal Society Open Science. It has now been reviewed for three reviewers and as you can see below the comments are largely

positive. There were however some issues/concerns that need to be addressed before acceptance can be considered.

Best wishes
Sean Murray

Reviewer comments to Author:

Reviewer: 1

Comments to the Author(s)

This, aNMJ-Morph macro, is an important improvement of an existing method to increase the time and congruence of analyzing NMJs. Despite the improvements, there is little mention of how the macro would perform with images of NMJs acquired using different scanning parameters, a critical consideration. There is also little mention if machine-learning algorithms could potentially outperform this macro and importantly be a more reliable and speedier method for assessing the morphology of NMJs taken using scanning parameters. Thus, the discussion should at least elaborate on this potential issues and future improvements.

Reviewer: 2

Comments to the Author(s)

The manuscript RSOS-200128 by Gavin Minty et al. describes the novel macro for semi-automated morphometric analysis of the confocal microscopy images of neuromuscular junctions. This macro-based analysis is a development of the manual workflow that this group published in 2016. The novel macro seems useful for the field, and the authors are making the software freely available. The manuscript is interesting and seems suitable for this journal; however, several points need to be addressed. Detailed review critiques are described below in the order of appearance and not by importance.

Page 3, lines 31 to 35, “aNMJ-morph conferred a 5-fold increase in data acquisition rate compared with the parent workflow, with average analysis times reduced to approximately 1 minute per NMJ” is an over-interpretation and needs revision.

- Page 7, lines 19 to 22, “average analysis time per NMJ reduced from nearly 5 and a half minutes (319 seconds) to just over 1 minute (79 seconds).” This improvement is only four-fold ($319/79=4.04$).
- The same issue is seen on page 14, lines 54 to 58 ($319/79=4.04$).
- The same issue applies to page 6, line 3, “the image acquisition time to \approx 1 minute per NMJ” seems like an overstatement of the difference.

Page 5, line 38.

Instructions should be given where to obtain the required plugin “the Binary Connectivity plugin” and how to install it.

Page 6, lines 53-57.

Has the macro been tested in both Windows or Macintosh platforms? If not, authors should specify which operating system has been used for testing the macro.

Page 8, Methods.

Describe the Fiji version number, the Fiji build number, and the manage update sites that are necessary to execute this workflow.

Page 8, lines 52 to 57. The Reviewer agrees to the authors for figure 2B being poor image quality. However, for figure 2C, the thresholding has been appropriately executed without evident noise in the image. Thus, the Reviewer disagrees with concluding this analysis problem as a result of

poor image quality. The authors need to investigate the cause further and how to deal with this kind of situation. Most importantly, the Reviewer disagrees with excluding NMJs like this image from the analysis.

Page 9, lines 23 to 25.

The authors should describe in detail about thresholding, whether it is appropriate to use the same or different thresholding for each NMJ image.

Page 9, line 38, "the were" should be corrected.

Page 10, lines 3 to 9, "NMJ images."

The authors must elaborate on the images used in this macro. What image file-type would be acceptable for the analysis? Which microscope manufacture original file type could be used directly in this workflow? Do the users need to load additional Fiji plugins to read specific file types necessary for the aNMJ-morph?

If not, specify which file type is compatible. The Reviewer assumes that the confocal Z-stack needs to be projected. If so, specify what type of projection is suitable for this analysis?

Page 14, line 11, "batch processing." Instruction seems to be missing for how to batch process images.

Page 14, lines 12 to 14, "The Reviewer assumes that the confocal Z-stack needs to be projected. If so, specify what type of projection is suitable for this analysis?"

Erasing the axon using a paintbrush is a manual input to the analysis. The same issue is seen on page 9, lines 19 to 23.

Reviewer: 3

Comments to the Author(s)

see attached file. (Reviwer X - OpenRoyalSocManuscript-2020.pdf)

Author's Response to Decision Letter for (RSOS-200128.R0)

See Appendix A.

Decision letter (RSOS-200128.R1)

23-Mar-2020

Dear Dr Jones,

It is a pleasure to accept your manuscript entitled "aNMJ-morph - A simple macro for rapid analysis of neuromuscular junction (NMJ) morphology" in its current form for publication in Royal Society Open Science.

You can expect to receive a proof of your article in the near future. Please contact the editorial office (openscience_proofs@royalsociety.org) and the production office (openscience@royalsociety.org) to let us know if you are likely to be away from e-mail contact -- if

you are going to be away, please nominate a co-author (if available) to manage the proofing process, and ensure they are copied into your email to the journal.

on behalf of Dr Sean Murray (Associate Editor) and Malcolm White (Subject Editor)
openscience@royalsociety.org

Appendix A

ROSS A. JONES
BSc, MSc, PhD, MBChB, MRCS, FAS
Lecturer in Clinical & Surgical Anatomy
*EDINBURGH MEDICAL SCHOOL:
BIOMEDICAL SCIENCES*
The University of Edinburgh
Old Medical School (Anatomy)
Teviot Place
Edinburgh EH8 9AG

Telephone: +44 (0)131 6515207
Email: Ross.Jones@ed.ac.uk

20 March 2020

Dear Dr Kristiansen

Re: Royal Society Open Science - Decision on Manuscript ID RSOS-200128

Thank you for your email of 16 March informing us that the above manuscript has been accepted for publication in Open Science, subject to minor revisions in accordance with the referees' suggestions. We are very grateful for the critical appraisal and valuable feedback that has been provided and welcome the opportunity to respond to the referees' comments.

We believe that our updated manuscript has now addressed all of the reviewers' remarks. Full details of the changes can be found in the specific responses to the referees' comments below. All changes to the manuscript have been highlighted in blue font and marked in grey. We hope that you find these revisions to be satisfactory.

Once again, we are delighted that our manuscript has been accepted for publication in Royal Society Open Science.

Yours faithfully

Ross A Jones
(on behalf of all co-authors)

Response from Authors (Open Science manuscript RSOS-200128)

We are very grateful to the referees for providing constructive feedback on the original manuscript. We believe that the manuscript has been improved by addressing the issues highlighted. Please find below a point-by-point response to each of the individual comments, along with details of changes and additions to the manuscript.

Responses to Reviewer Comments to the Author(s)

Reviewer: 1

This, aNMJ-morph macro, is an important improvement of an existing method to increase the time and congruence of analyzing NMJs. Despite the improvements, there is little mention of how the macro would perform with images of NMJs acquired using different scanning parameters, a critical consideration. There is also little mention if machine-learning algorithms could potentially outperform this macro and importantly be a more reliable and speedier method for assessing the morphology of NMJs taken using scanning parameters. Thus, the discussion should at least elaborate on this potential issue and future improvements.

Response: Many thanks indeed for the very positive comments. The points raised are valid and important, and we have included the following additional text in the Results and Discussion:

“We anticipate that other research groups will now wish to trial the macro in different settings, e.g. with NMJ images acquired using different scanning parameters and/or file types. To support these adaptations, we recommend that users first validate the macro output against equivalent data generated using the original workflow (6) to confirm the functionality of the macro in different settings. We also encourage the development of machine-learning algorithms based on the existing NMJ-morph approach to further refine and improve the rate of data acquisition.”

Reviewer: 2

The manuscript RSOS-200128 by Gavin Minty et al. describes the novel macro for semi-automated morphometric analysis of the confocal microscopy images of neuromuscular junctions. This macro-based analysis is a development of the manual workflow that this group published in 2016. The novel macro seems useful for the field, and the authors are making the software freely available. The manuscript is interesting and seems suitable for this journal; however, several points need to be addressed. Detailed review critiques are described below in the order of appearance and not by importance.

Response: Many thanks indeed for the very positive comments.

Page 3, lines 31 to 35, “aNMJ-morph conferred a 5-fold increase in data acquisition rate compared with the parent workflow, with average analysis times reduced to approximately 1 minute per NMJ” is an over-interpretation and needs revision.

Response: We agree that the value of 5-fold is misleading in relation to the values given on the bar chart in Figure 3 (and as noted in the comments below). We have therefore replaced “5-fold” with “4-fold” throughout the manuscript.

Page 7, lines 19 to 22, “average analysis time per NMJ reduced from nearly 5 and a half minutes (319 seconds) to just over 1 minute (79 seconds).” This improvement is only four-fold ($319/79=4.04$).

Response: As above – “5-fold” replaced with “4-fold”.

The same issue is seen on page 14, lines 54 to 58 (319/79=4.04).

Response: As above – “5-fold” replaced with “4-fold”.

The same issue applies to page 6, line 3, “the image acquisition time to \approx 1 minute per NMJ” seems like an overstatement of the difference.

Response: Following from above, we have now substituted the phrase “just under one and a half minutes”.

Page 5, line 38.

Instructions should be given where to obtain the required plugin “the Binary Connectivity plugin” and how to install it.

Response: We have now included an additional section in the Methods as follows:

“ImageJ/Fiji and Binary Connectivity

The macro was developed using ImageJ/Fiji software (version: 2.0.0-rc-67/1.52i / build: 1762a07c5c). The latest version of ImageJ/Fiji is freely available at <https://fiji.sc> including instructions for download. The Binary Connectivity plugin is freely available at <https://blog.bham.ac.uk/intellimic/g-landini-software> under the section ‘Morphological Operators for ImageJ’ (including instructions for installation). To manage updates, the latest version of the macro will be hosted at Edinburgh DataShare (13).”

Page 6, lines 53-57.

Has the macro been tested in both Windows or Macintosh platforms? If not, authors should specify which operating system has been used for testing the macro.

Response: We can confirm that the macro has been tested and functions on both Microsoft Windows and Apple Mac operating systems. We have therefore added the following text at the relevant points in the Results and Discussion:

“...compatible with both Windows and Mac operating systems...”

“...each investigator used a different workstation and operating system (to ensure compatibility with both Windows and Mac)...”

Page 8, Methods.

Describe the Fiji version number, the Fiji build number, and the manage update sites that are necessary to execute this workflow.

Response: We have now included an additional section in the Methods as follows:

“ImageJ/Fiji and Binary Connectivity

The macro was developed using ImageJ/Fiji software (version: 2.0.0-rc-67/1.52i / build: 1762a07c5c). The latest version of ImageJ/Fiji is freely available at <https://fiji.sc> including instructions for download. The Binary Connectivity plugin is freely available at <https://blog.bham.ac.uk/intellimic/g-landini-software> under the section ‘Morphological Operators for ImageJ’ (including instructions for installation). To manage updates, the latest version of the macro will be hosted at Edinburgh DataShare (13).”

Page 8, lines 52 to 57. The Reviewer agrees to the authors for figure 2B being poor image quality. However, for figure 2C, the thresholding has been appropriately executed without evident noise in the image. Thus, the Reviewer disagrees with concluding this analysis problem as a result of poor image quality. The authors need to investigate the cause further and how to deal with this kind of situation. Most importantly, the Reviewer disagrees with excluding NMJs like this image from the analysis.

Response: We also agree that the thresholding in 2C is appropriate for this particular image, which nevertheless segments in the abnormal manner depicted. Having excluded poor image

quality, we can only attribute the aberrant segmentation in this particular instance to a glitch in ImageJ. In our experience, this is extremely rare (<1 in 1,000 images). In these circumstances, variables related to segmentation (i.e. number of clusters and derivations) must be necessarily excluded (hence the ‘check segmentation’ window in the macro) – all other variable (e.g. areas, perimeters, etc.) can of course be measured and included. We have therefore added the following text to clarify:

“...In exceptional circumstances, aberrant segmentation is noted in images of sound quality (Figure 2C); in these instances, automated counting of clusters is not possible (measurement of other variables is unaffected, e.g. area, perimeter, etc.). To address these eventualities in the macro, an instruction window was incorporated prompting users to confirm appropriate segmentation of the image (Figure 1, window 6/7; Supplementary File 1); in circumstances of abnormal segmentation, the macro will still measure and record the other variables...”

Page 9, lines 23 to 25.

The authors should describe in detail about thresholding, whether it is appropriate to use the same or different thresholding for each NMJ image.

Response: We agree that accurate image thresholding is the most critical aspect of NMJ-morph and this is discussed extensively in the original manuscript (Jones et al, 2016). We have therefore added the following text in the Methods:

“...the original NMJ-morph manuscript (6) should be consulted for detailed instruction/discussion of image thresholding. Of note, thresholded binary images must be compared to the original raw images to confirm accurate image reproduction.”

Page 9, line 38, “the were” should be corrected.

Response: Thank you for noting – we have made this correction.

Page 10, lines 3 to 9, “NMJ images.”

The authors must elaborate on the images used in this macro. What image file-type would be acceptable for the analysis? Which microscope manufacture original file type could be used directly in this workflow? Do the users need to load additional Fiji plugins to read specific file types necessary for the aNMJ-morph?

AND

If not, specify which file type is compatible. The Reviewer assumes that the confocal Z-stack needs to be projected. If so, specify what type of projection is suitable for this analysis?

Response: (Both comments) Thank you for raising these important points. We have added the following text to the Methods:

“NMJ images and file types

All images were captured using Zeiss/Nikon confocal microscopes, with the file types .lsm/.nd2 respectively. The macro uses the maximum intensity projection of the corresponding z-stack, and has been validated for these file types and image formats only. For all other file types, we recommend that users first validate the macro output against equivalent data generated with the original workflow (6) before proceeding.”

Page 14, line 11, “batch processing.” Instruction seems to be missing for how to batch process images.

Response: We apologise for the omission and have added the following text to the Figure 1 Legend:

“Note: For single NMJ analysis, first open the image, then select the macro from the plugins. For batch processing, first open the macro, then select the image folder; the macro will automatically cycle through each image in turn to completion.”

Page 14, lines 12 to 14, “The Reviewer assumes that the confocal Z-stack needs to be projected. If so, specify what type of projection is suitable for this analysis?” Erasing the axon using a paintbrush is a manual input to the analysis. The same issue is seen on page 9, lines 19 to 23.

Response: Please see the above comments in relation to file types and image formats (additional text has now been added to the manuscript). We have also updated the relevant text in relation to axon processing:

(in Legends) “...The only manual inputs include image thresholding, axon processing (measure/erase) and confirmation of image segmentation...”

(in Methods) “...The only manual aspects of the original NMJ-morph workflow to be retained in the macro related to image thresholding and axon processing (Figure 1, windows 1-5/7)...”

Reviewer: 3

This is a methods paper aimed at improving the workflow in the morphological analyses of the neuromuscular synapse. Mapping the morphology of neuromuscular synapses that have altered in different species, and inferring such changes to the human neuromuscular synapse is a worthy challenge. This is because the shapes of neuromuscular synapses are vastly different across species (e.g. see papers by Clark Slater). Given this, it would be good to know how robust for the 19 NMJ variables how they can be adapted across species. I think the authors could easily demonstrated this – by perhaps showing some of their excellent comparative data from aged human and mouse NMJ – where they did employ these NMJ variables to compare and contrast mouse and human NMJs that they published in Cell Reports. For example, in figure 1 and 2 show the NMJs rodent (top panel) and human NMJs for the same/similar variable in the lower panel of each figure. Overall, I think is a very fine methods paper and should be of great value to those interested in assessing NMJ morphologies across a variety of pathophysiological conditions. I also as suggested by the text, visited the latest aNMJ-morph macro and the demo – trailed it for use – as this is the practical part of the paper.

Response: Many thanks indeed for the very positive comments. Trialling the macro across a range of mammalian species (e.g. mouse, human, etc.) and in different age groups is an excellent suggestion worthy of future consideration, but is beyond the scope of the present methods paper. Many thanks also for taking the time to trial the macro in a practical setting as part of the review process – we are very encouraged by the positive feedback.

Some possible suggestions for the authors to consider are:

1) When moving from a raw image to a binary image it might help to remind the user to threshold the background from the signal prior to creating a binary image.

Response: This is a good point. As per the original NMJ-morph guidelines, thresholding should always be performed with reference to a duplicate copy of the original image, and we have therefore added the following text to the Methods:

“...Of note, thresholded binary images must be compared to the original raw images to confirm accurate image reproduction.”

2) Might be helpful for the demo to include some instruction on how to move around the image – either remind the user to place the mouse cursor to the area they wanted to zoom in or the demo can remind users to use the scrolling tool.

Response: These are excellent suggestions that we will aim to include on future versions of the demo tutorial video.

3) Set escape at any step point so the user can exit the marco at any time.

Response: Apologies for any confusion - the macro can already be exited at any time by pressing escape (see hint in window 2/7 of macro). We will aim to make this point more explicit on future versions of the demo tutorial video.

4) It might be good to include some exception catching in the script to avoid problems or crashes. Researchers might make mistakes while they are operating the marco or ImageJ or even the image file itself might cause a problem. It is wise to catch those problems by setting up exception measures to make sure the inputs are appropriate. Users might not be able to spot an error on their own or they just simply misunderstand how to use aNMJ-morph or ImageJ.

Response: This is an excellent suggestion for future versions of the macro, but is beyond the scope of our programming/scripting expertise at the present time; we will aim to incorporate this functionality in future macro updates at Edinburgh DataShare. Similarly, the authors welcome any further comments and suggestions for improvements from users of the workflow/macro. A full understanding of NMJ-morph is a pre-requisite for use of the macro, especially in relation to error identification, and as stated in the manuscript "...we recommend that users of aNMJ-morph are familiar and competent with the use of the original workflow in a practical setting (6)."